# On the equivalence of molecular graph convolution and molecular wave function with poor basis set

**Masashi Tsubaki**
National Institute of Advanced Industrial Science and Technology
`tsubaki.masashi@aist.go.jp`

**Teruyasu Mizoguchi**
Institute of Industrial Science, University of Tokyo
`teru@iis.u-tokyo.ac.jp`

## Abstract

In this study, we demonstrate that the linear combination of atomic orbitals (LCAO), an approximation introduced by Pauling and Lennard-Jones in the 1920s, corresponds to graph convolutional networks (GCNs) for molecules. However, GCNs involve unnecessary nonlinearity and deep architecture. We also verify that molecular GCNs are based on a poor basis function set compared with the standard one used in theoretical calculations or quantum chemical simulations. From these observations, we describe the quantum deep field (QDF), a machine learning (ML) model based on an underlying quantum physics, in particular the density functional theory (DFT). We believe that the QDF model can be easily understood because it can be regarded as a single linear layer GCN. Moreover, it uses two vanilla feedforward neural networks to learn an energy functional and a Hohenberg–Kohn map that have nonlinearities inherent in quantum physics and the DFT. For molecular energy prediction tasks, we demonstrated the viability of an "extrapolation," in which we trained a QDF model with small molecules, tested it with large molecules, and achieved high extrapolation performance. We believe that we should move away from the competition of interpolation accuracy within benchmark datasets and evaluate ML models based on physics using an extrapolation setting; this will lead to reliable and practical applications, such as fast, large-scale molecular screening for discovering effective materials.

## 1 Introduction

Recently, graph convolutional networks (GCNs) [1, 2] have been applied to molecular graphs. Although numerous variants of the molecular GCN have been developed [3, 4, 5] (Section 2.1), they have a basic computational procedure: the GCN model (1) considers that each node (i.e., atom) of a molecular graph has a multidimensional variable (i.e., the atom feature vector), (2) uses the convolutional operation to update the feature vectors according to the graph structure defined by the adjacency or distance matrix between the atoms in the molecule, and finally (3) outputs a value (e.g., the energy of the molecule) via a readout function (e.g., the sum/mean of the updated vectors). Deep neural networks (DNNs) are used in the convolutional operation. Therefore, GCNs involve strong nonlinearity when modeling the molecular graph structure and have achieved good prediction performance on large-scale benchmark datasets, such as QM9 [6].

In this study, from the perspective of quantum physics, we demonstrate that extant molecular GCNs involve unnecessary nonlinearity and deep architecture. We first describe an approximation of quantum physics introduced by Pauling and Lennard-Jones in the 1920s [7, 8, 9], which states that the

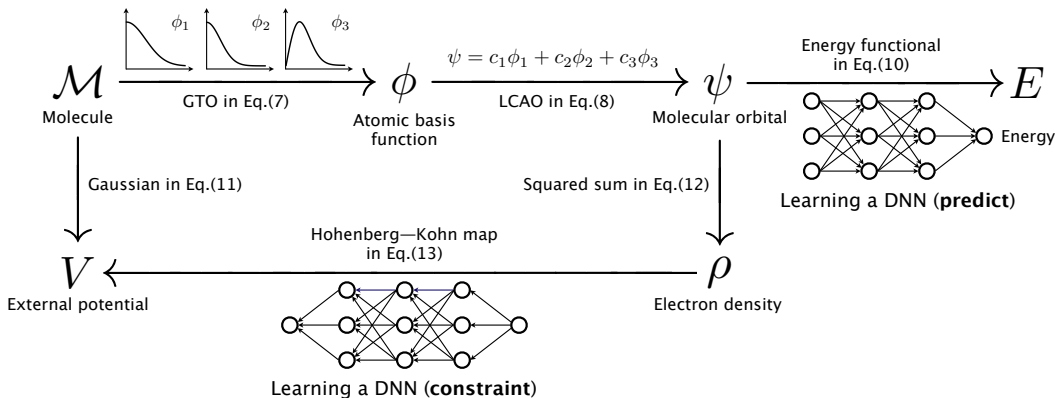

Figure 1: Overview of the computational graph of our proposed quantum deep field (QDF) framework from the input molecule $\mathcal{M}$ to the output energy $E$. The energy functional predicts $E$ and the Hohenberg–Kohn (HK) map imposes the potential constraint on the electron density $\rho$.

superposition of atomic wave functions (called orbitals) is based on their linear combination (Section 2.2). We demonstrate that this linear superposition/combination corresponds to the convolutional operation in GCNs; that is, the nonlinear DNN used in the above (2) is not required for modeling the molecular structure (Section 3). Additionally, in the linear superposition/combination, although the number of wave functions/orbitals (or basis functions) and the types of each basis function are important, the molecular GCNs do not consider these points. In particular, the reason for performance degradation in deeper GCNs has been discussed recently in [2, 10]; however, it is trivial with regard to molecules. The molecular GCNs are built on a poor and incorrect basis function set compared with the standard one used in theoretical calculations or quantum chemical simulations.

From these observations, we describe the quantum deep field (QDF), a machine learning (ML) model based on an underlying quantum physics, in particular the density functional theory (DFT) [11]. The model is separated into linear and nonlinear components. The former is the linear combination of atomic orbitals (LCAO) [7, 8, 9], which is implemented through matrix–vector multiplication; the latter is the energy functional that has nonlinearity inherent in quantum physics. This study implements this nonlinear functional using a vanilla feedforward DNN (Section 4.1). Additionally, over the entire model, we impose a physical constraint based on the Hohenberg–Kohn theorem [12], which has nonlinearity inherent in DFT and can therefore be implemented using a vanilla feedforward DNN (Section 4.2). The components and constraint can be represented as a computational graph that learns the energy in a supervised fashion (Figure 1), and all model parameters are trained by back-propagation and stochastic gradient descent (SGD) algorithms (Section 4.3). For atomization energy prediction with the QM9 dataset [6], our QDF model was competitive with a state-of-the-art model called SchNet [13] but with a million fewer parameters (Section 5.1).

Furthermore, this study demonstrated an "extrapolation" [14, 15] with regard to predicting the energies of totally unknown molecules, in which we trained a QDF model with small molecules, tested it with large molecules, and achieved high extrapolation performance (Section 5.2). In a standard ML evaluation, the training and test sets have the same data distribution; in other words, the molecular sizes and structures in both sets are the same or very similar. Under this "interpolation" evaluation, if a highly nonlinear DNN model is trained, it can easily fit to a physically meaningless but high-accuracy function that maps the input molecules into the output energies; this is because DNN can easily learn many non-linear properties inside the training data distribution [16, 17] unrelated with physics. However, the ML evaluation is mainly interested in the final output (i.e., interpolation accuracy within a benchmark dataset), and even if the energy prediction performs well, the model parameters may not always reflect physics. The extrapolation can evaluate ML models focusing on physical validity; this will lead to the development of more reliable and practical ML applications.

## 2 Background: molecular GCN and LCAO

**2.1 Molecular GCN.** A molecule is defined as $\mathcal{M} = \{(a_1, R_1), (a_2, R_2), \cdots, (a_M, R_M)\} = \{(a_m, R_m)\}_{m=1}^{M}$, where $a_m$ is the $m$th atom (e.g., H and O), $R_m$ is the 3D coordinate of $a_m$,

and $M$ is the number of atoms in $\mathcal{M}$. We consider a graph representation of $\mathcal{M}$, in which the node is $a_m$ and the edge between $a_m$ and $a_n$ is defined by the atomic distance $D_{mn} = ||R_m - R_n||$. In other words, the molecular graph $\mathcal{G}_{\mathcal{M}} = (V, D)$ is a fully connected graph, where $V$ is the set of atoms and $D \in \mathbb{R}^{M \times M}$ is the corresponding distance matrix.

Given $\mathcal{G}_{\mathcal{M}}$, we initialize each atom with a $d$-dimensional vector and denote the atom vector as $\mathbf{a}_m$, where $d$ is a hyperparameter. For example, $\mathbf{a}_m = [a_1, a_2, \cdots, a_d]$, in which each element is a feature value (e.g., the atomic charge, hybridization, and acceptor/donor) used in [3, 4, 5], or is randomly initialized and then learned via back-propagation and SGD [18, 13]. We then consider the convolutional operation to update $\mathbf{a}_m$ iteratively according to the graph structure $\mathcal{G}_{\mathcal{M}}$ as follows:

$$\mathbf{a}_m^{(\ell+1)} = \sum_{n=1}^{M} w(D_{mn}) \mathbf{h}_n^{(\ell)}, \tag{1}$$

where $\mathbf{a}_m^{(\ell+1)}$ is the $m$th atom vector in layer $\ell + 1$ (where layer means the number of updates or iterations), $\mathbf{h}_n^{(\ell)}$ is the hidden vector of $\mathbf{a}_n^{(\ell)}$ obtained by a neural network (e.g., $\mathbf{h}_n^{(\ell)} = \mathrm{ReLU}(\mathbf{W}^{(\ell)} \mathbf{a}_n^{(\ell)} + \mathbf{b}^{(\ell)}))$, and $w(D_{mn})$ is a function of the weight of $\mathbf{h}_n^{(\ell)}$ determined by the atomic distance. Various weight functions (or edge features) have been used; for example, the chemical bond type [3, 5], distance bin [4], inverse: $w(D_{mn}) = 1/D_{mn}$, Gaussian kernel transformation: $w(D_{mn}) = \exp(-\gamma ||D_{mn} - \mu||^2)$ [18, 13], and learnable vector: $w(D_{mn}) = \mathbf{u}_{mn} \in \mathbb{R}^d$ [19]. Note that, if we consider the molecular graph represented by the adjacency matrix $A$ instead of the distance matrix $D$, we have the binary weight $w(D_{mn}) = \{0, 1\}$ that corresponds to the bond.

**2.2 LCAO.** Herein, for the readers who are not familiar with quantum physics and chemistry, we start from a slightly incorrect but understandable description of LCAO (also called the linear superposition of wave functions) introduced by Pauling and Lennard-Jones in the 1920s [7, 8, 9]. We then provide a correct description to facilitate comparison of the equivalence and differences between molecular GCN and LCAO in Section 3.

We first consider that each atom $a_m$ has an inherent wave-like spread in 3D space centered on $R_m$; this is known as the electron cloud. As shown in Figure 2(a), we can consider this wave as a probability distribution. Based on the waves, we consider a value (i.e., the probability of electron) on an arbitrary position $r$ in the "field" of $\mathcal{M}$, not limited to the atomic position $R_m$, as follows:

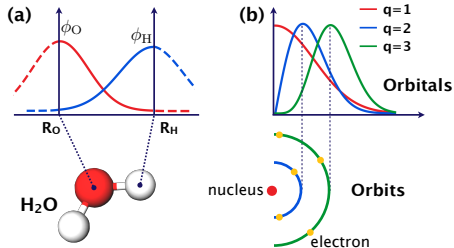

Figure 2: (a) Superposition between $\phi_O$ and $\phi_H$ corresponding to the chemical bond $O-H$ in $H_2O$. (b) "Orbitals" in a quantum view and "orbits" in a classical view.

$$\psi(r) = \sum_{m=1}^{M} c_m \phi_m(r - R_m), \tag{2}$$

where $\psi(r)$ is the value on $r$, $\phi_m(r - R_m)$ is the value on $r$ derived from the wave whose origin is $R_m$, and $c_m$ is the coefficient in this linear combination. In quantum physics or chemistry, we refer to $\phi$ as the atomic wave function or atomic orbital and $\psi$ as the molecular wave function or molecular orbital. For example, in a water molecule $H_2O$, the above linear combination can be expressed as follows:

$$\psi(r) = c_H \phi_H(r - R_H) + c_H \phi_H(r - R_{H'}) + c_O \phi_O(r - R_O). \tag{3}$$

Note that because the two hydrogen atoms in $H_2O$ have the same characteristics, its atomic orbital and coefficient are the same (i.e., $\phi_H$ and $c_H$), but we distinguish their different positions using $R_H$ and $R_{H'}$ in Eq. (3).

However, Eq. (3) is incorrect because the oxygen atom O has multiple electrons and we need to consider multiple atomic orbitals for O. That is, we rewrite the term O in Eq. (3) as follows:

$$\psi(r) = c_H \phi_H(r - R_H) + c_H \phi_H(r - R_{H'}) + \sum_k c_O^{(k)} \phi_O^{(k)}(r - R_O), \tag{4}$$

where $k$ is the number of atomic orbitals for the oxygen. Additionally, we have various choices for representing each atomic orbital in Eq. (4). Here, it is natural to represent an orbital using a large

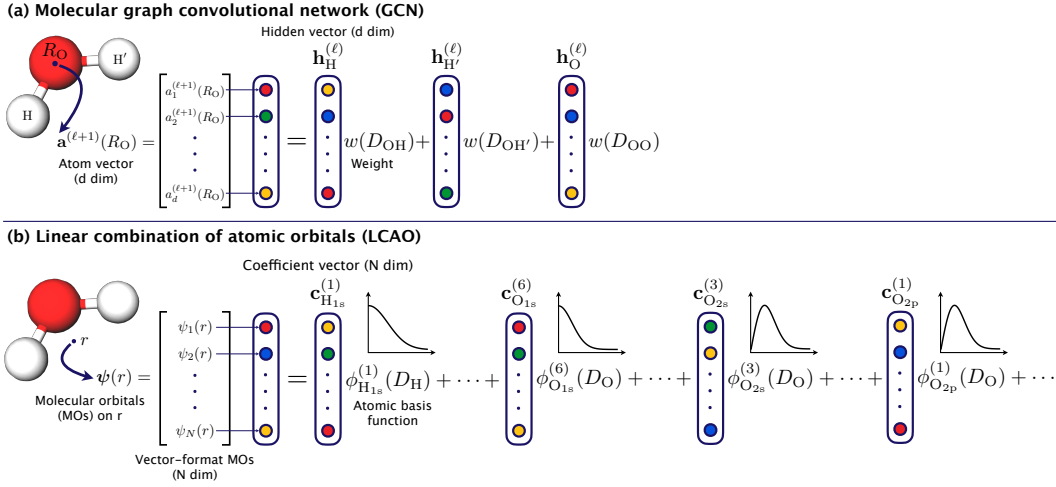

Figure 3: Both calculations involve the summation of the vector–scalar multiplications. (a) In the molecular GCN, the vectors are the atomic features and the scalars are their weights. (b) In the LCAO, the vectors are the coefficients and the scalars are the values of the atomic basis functions on $r$. Roughly, with a standard 6-31G basis set, we have six basis functions for the 1s orbital of the oxygen, which we denote as $\phi_{O_{1s}^{(1)}}, \phi_{O_{1s}^{(2)}}, \cdots, \phi_{O_{1s}^{(6)}}$. Additionally, we have four (i.e., $3 + 1 = 4$) basis functions for the 1s orbital of the hydrogen and the 2s/2p orbitals of the oxygen, which we denote as $\phi_{H_{1s}^{(1)}}, \cdots, \phi_{H_{1s}^{(4)}}, \phi_{O_{2s}^{(1)}}, \cdots, \phi_{O_{2s}^{(4)}}$, and $\phi_{O_{2p}^{(1)}}, \cdots, \phi_{O_{2p}^{(4)}}$. Each $N$-dimensional coefficient vector is also characterized by a subscript (e.g., $\mathbf{c}_{H_{1s}^{(1)}}, \mathbf{c}_{O_{1s}^{(6)}}, \mathbf{c}_{O_{2s}^{(3)}}$, and $\mathbf{c}_{O_{2p}^{(1)}}$).

number of known valid (e.g., Gaussian) basis functions. That is, we further rewrite Eq. (4) using $\phi^{(\cdot)}$, which is called the atomic basis function as follows:

$$\psi(r) = \sum_i c_H^{(i)} \phi_H^{(i)}(r - R_H) + \sum_i c_H^{(i)} \phi_H^{(i)}(r - R_{H'}) + \sum_j \sum_k c_O^{(j,k)} \phi_O^{(j,k)}(r - R_O). \qquad (5)$$

Thus, Eq. (5) has a nested structure, in which the molecular orbital $\psi(r)$ has multiple atomic orbitals $\phi_a(r - R_a)$ and each atomic orbital has multiple atomic basis functions $\phi_a^{(\cdot)}(r - R_a)$. Generally, we flatten this nested structure and describe it as follows:

$$\psi(r) = \sum_{n=1}^{N} c_n \phi_n(r - R_n) \quad \text{s.t.} \quad \sum_{n=1}^{N} c_n^2 = 1, \qquad (6)$$

where $N$ is the total number of basis functions and the coefficients are normalized. Eq. (3) has only three terms when $\mathcal{M} = H_2O$, whereas, Eq. (6) can have a large number of terms (in principle, infinite) for better approximating $\psi$. The set of basis functions (i.e., the basis set) determines the level of computational accuracy in theoretical calculations or quantum chemical simulations. For example, with a standard basis set, such as 6-31G [9] that is widely used in theoretical calculations, $H_2O$ has more than 30 basis functions (i.e., $N > 30$) and $C_6H_6$ (benzene) has more than 100 basis functions (i.e., $N > 100$).

In addition to the number of basis functions, the types of each basis function are important. The theoretical calculations often use the Gaussian-type orbital (GTO) as follows:

$$\phi_n(r - R_n) = \frac{1}{Z(q_n, \zeta_n)} D_n^{(q_n - 1)} e^{-\zeta_n D_n^2}, \qquad (7)$$

where $D_n = ||r - R_n||, q_n = 1, 2, \cdots$ is the principle quantum number, $\zeta_n$ is the control parameter of the Gaussian expansion (called the orbital exponent), and $Z(q_n, \zeta_n)$ is the normalization term. Note that this GTO is simplified in terms of the spherical harmonics. In Eq. (7), $D_n^{(q_n - 1)}$ allows the orbital to shift the peak of the Gaussian expansion, which corresponds to the classical "orbit" as shown in Figure 2(b). This is because the wave function is referred to as the "orbital."

We finally provide the correct description of LCAO. Indeed, the LCAO has $N$ multiple molecular orbitals and represents the linear combination using the coefficient vector $\mathbf{c}_n \in \mathbb{R}^N$ as follows:

$$\boldsymbol{\psi}(r) = \sum_{n=1}^{N} \mathbf{c}_n \phi_n(r - R_n), \qquad (8)$$

where $\boldsymbol{\psi}(r) \in \mathbb{R}^N$ is the vector-format molecular orbitals on $r$ and its $n$th element (i.e., $\psi_n(r)$) is the $n$th molecular orbital. Note that an initial assumption in LCAO is that the number of molecular orbitals is equal to the number of atomic orbitals (or basis functions). In other words, the vector dimensionality $N$ and the number of basis functions $N$ are the same. Thus, as we enhance the computational accuracy for approximating $\boldsymbol{\psi}(r)$ by increasing the number of basis functions, the dimensionality of $\boldsymbol{\psi}(r)$ also increases.

## 3 Equivalence and difference between molecular GCN and LCAO

Herein, in the molecular GCN, we rewrite the left side of Eq. (1) as $\mathbf{a}_m^{(\ell+1)} = \mathbf{a}^{(\ell+1)}(R_m)$ because the atom vector is defined on the atomic position $R_m$. Additionally, we transpose two terms $w(D_{mn})\mathbf{h}_n^{(\ell)}$ on the left side of Eq. (1) (i.e., the weight–vector) to $\mathbf{h}_n^{(\ell)}w(D_{mn})$ (i.e., the vector–weight). In the practical calculation of LCAO, we create a grid field of $\mathcal{M}$ and position $r$ in continuous space is regarded as a discrete point $r_i$ in the grid field (see Supplementary Material). Additionally, we rewrite the right side of Eq. (8) as $\phi_n(r_i - R_n) = \phi_n(D_{in})$ because the atomic basis function is a function of the distance $D_{in} = ||r_i - R_n||$, except for the other parameters $q_n$ and $\zeta_n$.

Therefore, Eq. (1) and Eq. (8) can be represented as follows:

$$\mathbf{a}^{(\ell+1)}(R_m) = \sum_{n=1}^{M} \mathbf{h}_n^{(\ell)}w(D_{mn}) \quad \text{and} \quad \boldsymbol{\psi}(r_i) = \sum_{n=1}^{N} \mathbf{c}_n\phi_n(D_{in}). \tag{9}$$

Thus, the two equations are easier to compare (Figure 3). In the following subsections, we find and discuss the equivalence and difference between the molecular GCN and LCAO.

**3.1 Atom vector $\mathbf{a}^{(\ell+1)}(R_m) \in \mathbb{R}^d$ and molecular orbital $\boldsymbol{\psi}(r_i) \in \mathbb{R}^N$.** On the left side of both equations in Eq. (9), the GCN has the $d$-dimensional vector on the atomic position $R$, and the LCAO has the $N$-dimensional vector on the field position $r_i$. We believe that this difference, in terms of the positions of the multidimensional variables, is not a serious problem in this case. We can assume $R$ as a representative point that the GCN considers. Actually, this is reasonable for modeling the molecule and its energy because the molecular energy can often be calculated as the summation of atomic energy contributions [20]. For example, in physical chemistry the neural network potential [20, 21, 14] based on the embedded atom method [22] has been proposed; this method considers the LCAO, molecular orbital, and electron density on the atomic position $R$, and not on the field position $r_i$ [23]. Furthermore, $d$ is a hyperparameter in the GCN and $N$ is the number of basis functions in the LCAO, which can be varied. When $d = N$, both the GCN and LCAO have information with the same expressive power on a position in 3D space.

**3.2 Hidden vector $\mathbf{h}_n^{(\ell)} \in \mathbb{R}^d$ and coefficient vector $\mathbf{c}_n \in \mathbb{R}^N$.** On the right side of both equations in Eq. (9), the GCN has the hidden (i.e., nonlinear transformed atom) vector $\mathbf{h}_n^{(\ell)} = \text{ReLU}(\mathbf{W}^{(\ell)}\mathbf{a}_n^{(\ell)} + \mathbf{b}^{(\ell)})$, which often contains various atomic features (e.g., the charge, hybridization, and acceptor/donor) in its elements [3, 4, 5]. On the other hand, the LCAO has the coefficient vector $\mathbf{c}_n$ and we can find that $\mathbf{h}_n^{(\ell)}$ corresponds to $\mathbf{c}_n$. However, $\mathbf{c}_n$ does not contain the atomic features and only has the normalization condition $\sum_{n=1}^{N} c_n^2 = 1$ in Eq. (6). Although $\mathbf{h}_n^{(\ell)}$ and $\mathbf{c}_n$ have such different characteristics, if both vectors are optimized for predicting/minimizing the energy of a molecule, their role is the same. Furthermore, we emphasize that the number of parameters and model complexity of the GCN are significantly greater than those of the LCAO, which is derived from $\mathbf{W}^{(\ell)}$, a nonlinear activation ReLU, and a $\ell$ times iterative procedure in the GCN.

**3.3 Weight $w(D_{mn}) \in \mathbb{R}$ and basis function $\phi_n(D_{in}) \in \mathbb{R}$.** In Eq. (9), we have the weight $w(D_{mn})$ on the hidden vector in the GCN and find that $w(D_{mn})$ corresponds to the basis function $\phi_n(D_{in})$ in the LCAO. We emphasize that the $d$-dimensional hidden vectors are summed and weighted by the distances in the GCN; in contrast, the basis functions by the distances are combined linearly with the $d$-dimensional coefficient vectors in the LCAO. For example, in a GCN model, if the weight function is a Gaussian kernel transformation (e.g., $w(D_{mn}) = \exp((D_{mn} - \mu)^2/\sigma^2)$), it has similar characteristics to the GTO; however, this does not include the term $D_{in}^{(q_n-1)}$ in Eq. (7). Therefore, this does not satisfy the original symmetry and cannot represent the different peaks in Figure 2(b). In particular, when a molecule is represented as a discrete graph using an adjacency matrix (i.e., $w(D_{mn}) = \{0, 1\}$), the atomic basis function (or orbital/wave function) is 0 or 1.

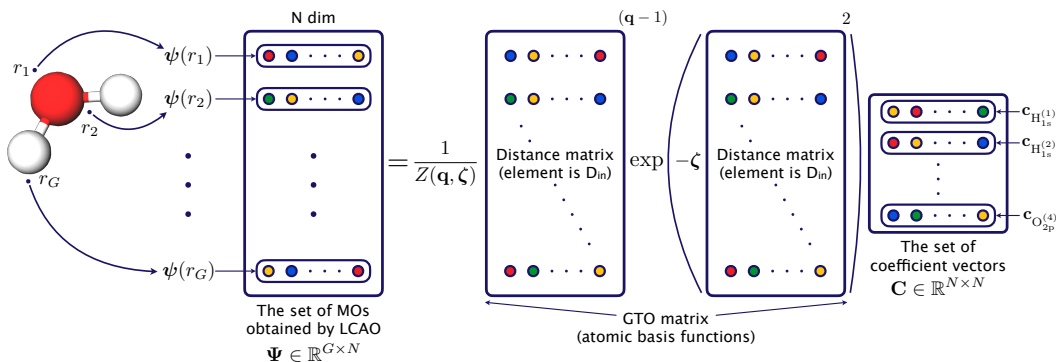

Figure 4: Intuitive explanation of the linear combination of atomic orbitals (LCAO) using batch-processing based matrix–vector multiplication (e.g., in PyTorch). The set of molecular orbitals on all points in the gird field, $\boldsymbol{\Psi} = \{\boldsymbol{\psi}(r_i)\}_{i=1}^{G}$, is efficiently obtained (i.e., processed as a batch on a GPU) by matrix-vector multiplication between the Gaussian-type orbital (GTO) matrix (i.e., each element is the atomic basis function $\phi(D_{in})$) and the set of coefficient vectors $\mathbf{C}$.

**3.4 Number of atoms $M$ and number of basis functions $N$.** Let us focus on the number of terms in Eq. (9). We can find that the GCN has $M$ number of terms (atoms), where, for example, $M = 3$ when $\mathcal{M} = $ H$_2$O. In contrast, the LCAO has $N$ number of terms (basis functions), where, for example, $N > 30$ when $\mathcal{M} = $ H$_2$O using a standard basis set (e.g., 6-31G, as mentioned in Section 2.2). The difference is more than 10 times; that is, the GCN has a very poor basis set. Recent studies have pointed out that GCNs cannot improve their performance by increasing the number of convolutional layers [2, 10]. While we are not familiar with GCN applications for other kinds of graph data, such as social, biological, and financial networks, for the molecular GCN and its variants, the above problem is trivial: the number of basis functions is insufficient and the type of each basis function is incorrect. The overparameterized, deeply hierarchical, and highly nonlinear GCN model builds on such poor and incorrect basis sets. Note that while the molecular GCN uses the $M \times M$ distance matrix, the LCAO uses the $G \times N$ GTO matrix, where $N$ is larger than $M$ and $G$ is much larger than $M$ (Figure 4). This is inevitable in the LCAO but a computational drawback (in particular, for the memory) compared with the GCN.

**3.5 Dimensionality of the atom vector $d$ and number of basis functions $N$.** The dimensionality $d$ of the atom vector in the GCN corresponds to the number of atomic features; the dimensionality $N$ of the coefficient vector in the LCAO corresponds to the number of basis functions, which determines the level of computational accuracy in theoretical calculations or quantum chemical simulations. Considering this along with Section 3.4, although it seems to enhance the expressive power of the GCN by increasing $d$, this is not physically meaningful owing to the fixed $M$, where $M$ is both the molecular size and the number of basis functions. For example, even if we use the dimensionality $d = 1000$ when $\mathcal{M} = $ H$_2$O, the number of basis functions is still only three in the GCN. In contrast, in LCAO, $N$ is not only the dimensionality of coefficient vector but also the number of basis functions. Therefore, increasing $N$ is physically meaningful for approximating $\boldsymbol{\psi}(r_i)$ and improving the computational accuracy. Furthermore, $N$ is usually determined by a basis set; therefore, we can assume that $N$ is an automatically determined value and not a hyperparameter, like in ML models.

A molecular GCN is regarded as a neural message passing algorithm [4], in which the atom is an object (i.e., a node) that has a message (i.e., a feature vector), and the atom messages are passed through the molecular graph structure using the adjacency or distance matrix in a nonlinear deep fashion; however, this is incorrect. More precisely, the atomic orbital or wave function is represented by the basis functions, and their linear combination or superposition is calculated using the multidimensional coefficients; this is the LCAO (see again Figure 3). Table 1 summarizes the characteristics of the molecular GCN and LCAO discussed in this section.

# 4 Learning energy functional and imposing physical constraint

**4.1 Energy functional.** Thus far, we have described the linearity in obtaining molecular orbitals from atomic orbitals. However, the relationship between molecular orbitals and energy has non-linearity inherent in quantum physics. Here, we use a neural network to estimate this nonlinear

relationship, which is called an energy functional (i.e., function $\mathcal{F}$ of function $f(x)$, $\mathcal{F}[f(x)]$) because the energy $E$ is a function of $\psi$ and $\psi$ is a function of $r$.

As shown in Figure 4, we have the set of $N$-dimensional vector-format molecular orbitals: $\mathbf{\Psi} = \{\boldsymbol{\psi}(r_i)\}_{i=1}^{G}$. We consider $\mathcal{F}_{\text{DNN}}$, which is a DNN-based energy functional as follows:

$$E'_{\mathcal{M}} = \mathcal{F}_{\text{DNN}}[\mathbf{\Psi}], \tag{10}$$

where $E'_{\mathcal{M}}$ is the predicted energy of $\mathcal{M}$. This study uses a vanilla feedforward DNN for the implementation of $\mathcal{F}_{\text{DNN}}$ (see Supplementary Material). We finally minimize the loss function: $\mathcal{L}_E = ||E_{\mathcal{M}} - E'_{\mathcal{M}}||^2$, where $E_{\mathcal{M}}$ is the actual energy provided by the training dataset.

**4.2 Physical constraint based on the Hohenberg–Kohn theorem.** Unfortunately, only minimizing $\mathcal{L}_E$ does not result in a physically meaningful learning model. Since the DNN-based energy functional has strong nonlinearity, $\mathcal{F}_{\text{DNN}}$ may output the actual energy even if its input molecular orbitals $\mathbf{\Psi}$ are incorrect. To address this, we impose a physical constraint in learning the model based on the Hohenberg–Kohn theorem [12], which ensures that "the external potential $V(r)$ is a unique functional of the electron density $\rho(r)$." $V(r)$ can be determined by the atomic charges $Z_m$ (e.g., $Z_{\text{H}} = 1$ and $Z_{\text{O}} = 8$) and their positions $R_m$. Additionally, $\rho(r)$ can be obtained by[1] $\rho(r) = \sum_{n=1}^{N} |\psi_n(r)|^2$. Furthermore, the aforementioned statement indicates that a relationship between $V(r)$ and $\rho(r)$ has a nonlinearity but one-to-one correspondence inherent in DFT. Here, we also use a neural network to estimate this nonlinear relationship; this is called a Hohenberg–Kohn (HK) map [24, 25]. The HK map works as the constraint on $\psi(r)$, which is the input of $\mathcal{F}_{\text{DNN}}$, and results in a physically meaningful learning model as a whole (see again Figure 1).

Formally, we first consider a Gaussian-based external potential [26, 24] on $r_i$ as follows:

$$V_{\mathcal{M}}(r_i) = -\sum_{m=1}^{M} Z_m e^{-||r_i - R_m||^2}. \tag{11}$$

Thus, this is the Gaussian expansions of atomic charges in 3D space. Note that the model assumes $V_{\mathcal{M}}(r_i)$ to be the correct external potential of $\mathcal{M}$; that is, $V_{\mathcal{M}}(r_i)$ is used as a target for minimizing loss in the model. Additionally, we have the electron density on $r_i$ as

$$\rho(r_i) = \sum_{n=1}^{N} |\psi_n(r_i)|^2 \tag{12}$$

and consider $\mathcal{HK}_{\text{DNN}}$, which is a DNN-based HK map as follows:

$$V'_{\mathcal{M}}(r_i) = \mathcal{HK}_{\text{DNN}}(\rho(r_i)), \tag{13}$$

where $V'_{\mathcal{M}}(r_i)$ is the predicted external potential of $\mathcal{M}$. This study uses a vanilla feedforward DNN for the implementation of $\mathcal{HK}_{\text{DNN}}$ (see Supplementary Material). We finally minimize the loss function: $\mathcal{L}_V = ||V_{\mathcal{M}}(r_i) - V'_{\mathcal{M}}(r_i)||^2$.

**4.3 Learning.** As a total learning model, we minimize $\mathcal{L}_E$ for predicting $E$ and $\mathcal{L}_V$ for imposing the potential constraint on $\rho$ "alternately." We believe that this is similar to the learning strategy of generative adversarial networks (GANs) [27]. The entire model optimizes all parameters in the LCAO, $\mathcal{F}_{\text{DNN}}$, and $\mathcal{HK}_{\text{DNN}}$ using only the back-propagation and SGD algorithms. This is our proposed QDF framework and we describe its optimization details in Supplementary Material.

It is important to note that QDF must consider the following physical condition:

$$N_{\text{elec}} = \int \rho(r) dr = \int \sum_{n=1}^{N} |\psi_n(r)|^2 dr \approx \sum_{i=1}^{G} \sum_{n=1}^{N} |\psi_n(r_i)|^2, \tag{14}$$

where $N_{\text{elec}}$ is the total electrons in $\mathcal{M}$ (e.g., when $\mathcal{M} = H_2O$, $N_{\text{elec}} = 10$). In other words, we must "keep" the total electrons in learning/updating the molecular orbitals with the iterative SGD algorithm. We implement this by transforming $\boldsymbol{\psi}_n$ in the SGD as follows:

$$\boldsymbol{\psi}_n \leftarrow \sqrt{\frac{N_{\text{elec}}}{N}} \frac{\boldsymbol{\psi}_n}{|\boldsymbol{\psi}_n|}. \tag{15}$$

Note that $\boldsymbol{\psi}_n$ is not the $n$th row vector but the $n$th column vector of matrix $\mathbf{\Psi}$ described in Figure 4. The model requires other physical normalizations in learning (see Supplementary Material).

| | GCN | LCAO |
|---|---|---|
| # of terms | $M$ (fixed) | $N \sim \infty$ |
| weight/basis | $w(D)$ | $D^{(q-1)} e^{-\zeta D^2}$ |
| vector dim | $d$ | $N \sim \infty$ |
| parameters | $\mathbf{W}^{(\ell)}, \mathbf{b}^{(\ell)}, \mathbf{a}$ | $\mathbf{c}$ |
| nonlinearity | e.g., ReLU | nothing |

Table 1: Characteristics of the molecular graph convolutional network (GCN) and the linear combination of atomic orbitals (LCAO).

| Model | Size | MAE |
|---|---|---|
| GCN | 483,631 | 1.58 |
| DTNN [18] | — | 1.51 |
| SchNet [13] | 1,676,133 | 1.23 |
| QDF | 495,262 | 1.21 |
| Chemical accuracy | | 1.00 |

Table 2: Model size and mean absolute error (MAE; kcal/mol) on the QM9under14atoms dataset.

## 5 Evaluation: prediction and extrapolation

**5.1 Energy prediction.** We first describe the prediction performance for the atomization energy at 0 K of the QM9under14atoms dataset, which is a subset of the QM9 dataset [6] (the dataset and training details are given in Supplementary Material). Table 2 shows the model sizes and final prediction errors (mean absolute error (MAE), lower is better) of the baseline GCN, proposed QDF, and other methods [18, 13] as references. SchNet, which is a variant of the deep tensor neural network (DTNN) proposed earlier, is a standard state-of-the-art deep learning model. We argue that the GCN is not a weak baseline because it achieves a reasonable performance that is similar to that of the DTNN. Additionally, we believe that the QDF outperforms or competes with SchNet in terms of the prediction error. However, in terms of the model size, SchNet has more than 1.5 million learning parameters. In contrast, QDF, which has less than half a million parameters, is much more compact. We note that these results can vary with the careful tuning of its hyperparameters and SchNet may outperform QDF; however, our main aim in this study is not to build a competitive model with regard to "interpolation" performance within a single benchmark dataset.

**5.2 Energy extrapolation.** Using a more practical evaluation setting, we demonstrate that the QDF can capture the physically meaningful energy functional. In physics, we can assume that if an ML model could capture fundamental quantum characteristics (i.e., the orbital/wave function $\psi$ and electron density $\rho$), the model would be able to conduct a prediction for totally unknown molecules. In other words, the QDF would be able to perform an "extrapolation" in predicting the energy of totally different sized and structured molecules. To substantiate this, we trained a model with small molecules consisting of fewer than 14 atoms (QM9under14atoms) and tested it with large molecules consisting of more than 15 atoms (QM9over15atoms) in the QM9 dataset (Figure 5(a)). Additionally, we used three kinds of energy properties provided by the QM9 dataset: the atomization energy at 0 K, zero point vibrational energy, and enthalpy at 298.15 K. As shown in Figure 5(b), (c), and (d), in each energy prediction, the GCN achieved accuracy comparable (or superior) to that of the QDF in interpolation; however, the QDF could maintain this accuracy even when the molecular size increased, whereas the GCN could not. We believe that these low MAEs in predicting unknown large molecules are evidence that the QDF can capture the physically meaningful energy functional and the fundamental quantum characteristics of molecules.

However, we also believe that the extrapolation performance for the atomization energy and enthalpy have room to improve. The current QDF uses the simplified GTO ignoring the spherical harmonics [15], simplified Gaussian external potential [24] that cannot reproduce the potential and density close to the nucleus, and two vanilla feedforward DNNs. To improve extrapolation performance, improvements to these factors (e.g., using Slater-type orbitals (STOs), not GTOs) will be required.

In terms of computational cost, although the LCAO requires the $G \times N$ matrix described in Figure 4 and Section 3.4, training a QDF model using 10k molecules of the QM9under14atoms dataset can be done within 6 hours on a standard single (e.g., GTX 1080Ti) GPU. Once the model is trained, the prediction for 100k molecules of the QM9over15atoms dataset can be done within a few minutes.

## 6 Discussion, conclusion, and future directions

Our current implementation of the QDF model has some limitations. In this implementation, orbital exponents $\{\zeta_n\}_{n=1}^N$ and coefficient vectors $\{\mathbf{c}_n\}_{n=1}^N$ in the LCAO are the global common learning parameters for all molecules (i.e., are not specific to each molecular structure). These parameters

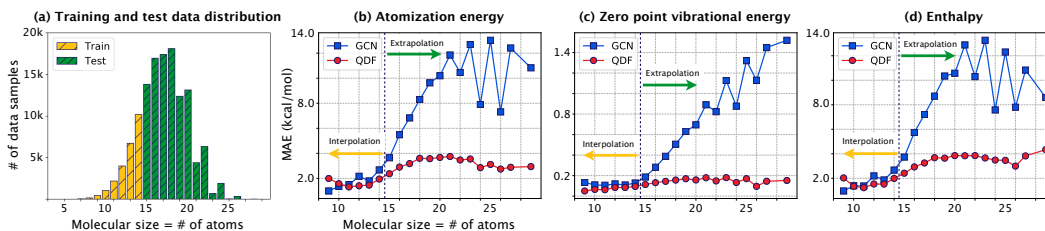

Figure 5: (a) Data distribution of training (interpolation) and test (extrapolation) samples. The number of test samples of the QM9over15atoms dataset is 115k molecules, which is 10 times more than that of training samples of the QM9under14atoms dataset. (b), (c), and (d) are the mean absolute errors (MAEs) of the atomization energy, zero point vibrational energy, and enthalpy of this large-scale extrapolation, respectively.

actually differ for each real molecule in theoretical calculations or quantum chemical simulations, and this is considered in standard GCNs via iterative nonlinear convolutional operations. However, we believe that our global parameters and the linearity prevent the model from being too flexible for each molecule in the training dataset; this is a general and practical trade-off problem in optimizing a model. Considering this trade-off, the current implementation of QDF is reasonable in terms of reducing the model parameters and complexity, resulting in robust extrapolation. Additionally, $N$ is also different for each real molecule; however, ML models need to set a global $N$ common for all molecules in a dataset. Indeed, the QM9 dataset includes only small organic molecules comprising H, C, N, O, and F atoms, and the maximum molecular size $M$ is 29. Considering these and a standard basis set called 6-31G, we set $N = 200$ as a valid value in theoretical calculations (for other hyperparameters, see Supplementary Material). Furthermore, this study considers an extrapolation evaluation setting in terms of the number of atoms; however, some studies have demonstrated extrapolation in terms of heavy atoms [14, 15]. It is important to design various extrapolations using various much larger datasets (e.g., the ANI-1 [14] and Alchemy [28]), which will require more efficient implementation and learning of QDF with dozens of GPUs, in the future.

This study described the QDF framework, which is different from other graph-based deep learning approaches for molecules; in other words, QDF is not an extension of existing molecular GCN and SchNet models. Our aim was to design a simple ML model consistently based on physics without incorporating complicated DNN techniques/architectures. Recently, some ML approaches have been proposed [29, 15, 23, 30, 31] as supervised learning models for the orbital/wave function $\psi$ and electron density $\rho$. In contrast, our QDF can be viewed as an unsupervised model to reproduce $\rho$, originated from $\psi$, using only a large-scale dataset of energy properties; other training strategies, not the current GAN-like one (Section 4.3) that may be unstable, will be considered in the future.

More generally, QDF can be viewed as one of the approaches such as the physics-informed, Hamiltonian, Fermionic, and Pauli neural networks [32, 33, 34, 35, 36, 37, 38]; these solve the physical problems and equations using physically meaningful modeling. QDF is designed as a self-consistent learning machine for DFT or to solve the Kohn–Sham equations [11] with minimal (three) components: LCAO, $\mathcal{F}_{DNN}$, and $\mathcal{HK}_{DNN}$. We believe that integrating a supervised model with a dataset of the electron density [39] (i.e., $\rho$ in Figure. 1 is given as a target) and an unsupervised but physically informed and meaningful model with a dataset of the atomization energy, HOMO–LUMO gap, and other properties [28] will yield an interesting hybrid ML model.

In the future, QDF will allow extensions for not only molecules but also for crystals [40, 19, 41] and other practical applications in materials informatics. In particular, the viability of extrapolation will lead to the development of applications with transfer learning for polymers [42], catalysts [43], photovoltaic cells [44, 45], and fast, large-scale screening for discovering effective materials.

**Code availability.** Our implementation is available at `https://github.com/masashitsubaki`. This provides the pre-trained model and model extensions with other datasets can be created.

## Broader Impact

This study will provide benefit for ML researchers who are interested in quantum physics/chemistry and applications for materials science/informatics.

## Acknowledgments and Disclosure of Funding

This study was supported by the Grant-in-Aid for Early-Career Scientists (Grant No. 20K19876) from the JSPS and the Grant-in-Aid for Scientific Research (Grant No. 19H05787 and 19H00818) from the MEXT.

## Footnotes

[1]This density is based on the one electron approximation. Precisely, $\boldsymbol{\psi}(r)$ is not the true molecular orbital and is called the Kohn–Sham orbital [11], which is a molecular orbital in a fictitious system of non-interacting electrons that provides the same density as the system of interacting electrons.

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
