[Supplementary Material]

# Supplementary Material:
# On the equivalence of molecular graph convolution and molecular wave function with poor basis set

**Masashi Tsubaki**
National Institute of Advanced Industrial Science and Technology
tsubaki.masashi@aist.go.jp

**Teruyasu Mizoguchi**
Institute of Industrial Science, University of Tokyo
teru@iis.u-tokyo.ac.jp

**Dataset.** The QM9 dataset [1] contains approximately 130,000 molecules made up of H, C, N, O, and F atoms along with 13 quantum chemical properties (e.g., atomization energy, HOMO, and LUMO) for each molecule. These molecular properties were calculated using a hybrid quantum simulation (Gaussian 09) at the B3LYP/6-31G(2df,p) level of theory. In this study, we created a subset of the QM9 dataset with a limited number of atoms, $M \leqq 14$, per molecule, which we refer to as the "QM9under14atoms" dataset in the main text. As the learning/predicting targets, we selected three kinds of energy properties: atomization energy at 0 K, zero point vibrational energy, and enthalpy at 298.15 K. The number of data samples in the QM9under14atoms dataset is approximately 15,000 molecules and we randomly shuffled and split this dataset into training, development (or validation), and test sets with a ratio of 8:1:1, in which the development set was used to tune the model and optimization hyperparameters. For large-scale extrapolation evaluation, we tested the trained QDF model using the "QM9over15atoms" dataset (i.e., $M \geqq 15$), in which the number of data samples was approximately 115,000 molecules.

**Molecular field definition.** Given a molecule $\mathcal{M} = \{(a_m, R_m)\}_{m=1}^{M}$, we consider spheres with a radius of $s$ Å, where each sphere covers each atom centered on $R_m$, and then divide the sphere into grids (or meshes) in intervals of $g$ Å. These $s$ and $g$ are the hyperparameters and this process yields many grid points in $\mathcal{M}$. The grid-based field of $\mathcal{M}$ is denoted by $\{r_1, r_2, \cdots, r_G\} = \{r_i\}_{i=1}^{G}$, where $r_i$ is the 3D coordinate of the $i$th point and $G$ is the total number of points.

**Architectures of $\mathcal{F}_{\mathbf{DNN}}$ and $\mathcal{HK}_{\mathbf{DNN}}$.** To implement the DNN-based energy functional $\mathcal{F}_{\text{DNN}}$ in the main text, we consider the following vanilla feedforward architecture:

$$\boldsymbol{\psi}^{(\ell+1)}(r_i) = \text{ReLU}(\mathbf{W}_E^{(\ell)} \boldsymbol{\psi}^{(\ell)}(r_i) + \mathbf{b}_E^{(\ell)}), \tag{1}$$

where $\ell = 1, 2, \cdots, L$ is the number of hidden layers ($\boldsymbol{\psi}^{(1)}(r_i) = \boldsymbol{\psi}(r_i)$ and $L$ is the final layer), ReLU is the nonlinear activation function, $\mathbf{W}_E^{(\ell)} \in \mathbb{R}^{N \times N}$ is the weight matrix in layer $\ell$, and $\mathbf{b}_E^{(\ell)} \in \mathbb{R}^N$ is the bias vector in layer $\ell$. We then sum over $\{\boldsymbol{\psi}^{(L)}(r_i)\}_{i=1}^{G}$ in the final layer $L$ and output an energy (i.e., atomization energy, zero point vibrational energy, or enthalpy) with the following vanilla linear regressor:

$$E'_{\mathcal{M}} = \mathbf{w}_E^{\top} \Big( \sum_{i=1}^{G} \boldsymbol{\psi}^{(L)}(r_i) \Big) + b_E, \tag{2}$$

where $\mathbf{w}_E \in \mathbb{R}^N$ is the weight vector and $b_E \in \mathbb{R}$ is the bias scalar. Figure 1 illustrates the architecture of this DNN-based energy functional.

Using a similar feedforward architecture, the DNN-based HK map $\mathcal{HK}_{\text{DNN}}$ is given by:

$$\mathbf{h}^{(1)}(r_i) = \mathbf{w}_\rho \rho(r_i) + b_\rho, \tag{3}$$

$$\mathbf{h}^{(\ell+1)}(r_i) = \text{ReLU}(\mathbf{W}_{\text{HK}}^{(\ell)}\mathbf{h}^{(\ell)}(r_i) + \mathbf{b}_{\text{HK}}^{(\ell)}), \tag{4}$$

$$V'_{\mathcal{M}}(r_i) = \mathbf{w}_V^\top \mathbf{h}^{(L')}(r_i) + b_V. \tag{5}$$

In this $\mathcal{HK}_{\text{DNN}}$, each hidden layer is $\mathbf{h} \in \mathbb{R}^{N'}$, where $N'$ is a hyperparameter. Figure 2 illustrates the architecture of this DNN-based HK map.

**Optimization.** Using back-propagation and an SGD (in practice, this study used the Adam optimizer [2]), we minimize the loss function $\mathcal{L}_E$ in the main text; in other words, we update the set of learning parameters $\Theta_E = \{\{\zeta_n\}_{n=1}^N, \{\mathbf{c}_n\}_{n=1}^N, \{\mathbf{W}_E^{(\ell)}\}_{\ell=1}^L, \{\mathbf{b}_E^{(\ell)}\}_{\ell=1}^L, \mathbf{w}_E, b_E\}$ as follows:

$$\Theta_E \leftarrow \Theta_E - \alpha \frac{1}{B} \sum_{k=1}^B \frac{\partial \mathcal{L}_{E_{\mathcal{M}_k}}}{\partial \Theta_E}, \tag{6}$$

where $\mathcal{L}_{E_{\mathcal{M}_k}}$ is the energy loss value of the $k$th molecule $\mathcal{M}$ in the training dataset, $\alpha$ is the learning rate, and $B$ is the batch size. Additionally, we also minimize the loss function $\mathcal{L}_V = \sum_{i=1}^G ||V_{\mathcal{M}}(r_i) - V'_{\mathcal{M}}(r_i)||^2$, i.e., we update the set of learning parameters $\Theta_V = \{\{\zeta_n\}_{n=1}^N, \{\mathbf{c}_n\}_{n=1}^N, \mathbf{w}_\rho, b_\rho, \{\mathbf{W}_{\text{HK}}^{(\ell)}\}_{\ell=1}^{L'}, \{\mathbf{b}_{\text{HK}}^{(\ell)}\}_{\ell=1}^{L'}, \mathbf{w}_V, b_V\}$ as follows:

$$\Theta_V \leftarrow \Theta_V - \alpha \frac{1}{B} \sum_{k=1}^B \frac{\partial \mathcal{L}_{V_{\mathcal{M}_k}}}{\partial \Theta_V}, \tag{7}$$

where $\mathcal{L}_{V_{\mathcal{M}_k}}$ is the potential loss value of the $k$th molecule $\mathcal{M}$ in the training dataset. We emphasize that QDF updates $\Theta_E$ and $\Theta_V$ "alternately" and note that the learning parameters in the LCAO, i.e., $\{\zeta_n\}_{n=1}^N$ and $\{\mathbf{c}_n\}_{n=1}^N$, are "shared" in $\Theta_E$ and $\Theta_V$.

**Normalization.** The LCAO considers the normalization for the coefficients in Eq. (6) in the main text. This can be implemented by updating the coefficients in the iterative SGD as follows:

$$\mathbf{c}'_n \leftarrow \frac{\mathbf{c}'_n}{|\mathbf{c}'_n|}. \tag{8}$$

Note that $\mathbf{c}'_n$ is not the $n$th row vector but the $n$th column vector of matrix $\mathbf{C}$ described in Figure 4 in the main text. Additionally, the normalization term in Eq. (7) in the main text is calculated as follows:

$$Z(q_n, \zeta_n) = \int |D_n^{(q_n-1)} e^{-\zeta_n D_n^2}|^2 dD = \sqrt{\frac{(2q_n - 3)!! \sqrt{\pi/2}}{2^{2(q_n-1)} \zeta_n^{(2q_n-1)/2}}}. \tag{9}$$

Note that because each Gaussian expansion $\zeta_n$ is a learning parameter of the model, $Z(q_n, \zeta_n)$ is recalculated every time the model parameters are updated in SGD.

**Hyperparameters.** All model and optimization hyperparameters and their values used in this study are listed in Table 1.

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

Figure 1: Architecture of our DNN-based energy functional $\mathcal{F}_{\mathrm{DNN}}$.

Figure 2: Architecture of our DNN-based Hohenberg–Kohn (HK) map $\mathcal{HK}_{\mathrm{DNN}}$.

| Hyperparameter | Value |
|---|---|
| Sphere radius $s$ | 0.75 Å |
| Grid interval $g$ | 0.3 Å |
| # of dimensions $N$ | 200 |
| # of hidden layers in $\mathcal{F}_{\text{DNN}}$ | 3 or 6 |
| # of hidden units in $\mathcal{HK}_{\text{DNN}}$ | 200 |
| # of hidden layers in $\mathcal{HK}_{\text{DNN}}$ | 3 or 6 |
| Batch size | 4, 16 |
| Learning rate | 1e-4 or 5e-4 |
| Decay of learning rate | 0.5 |
| Step size of decay | 200 or 500 epochs |
| Iteration | 1000 or 3000 epochs |

Table 1: List of all model and optimization hyperparameters and their values.