[Reviews · NeurIPS 2020]

Review 1

Summary and Contributions: This paper makes three contributions: 1. An opinion piece that can be summarized as a call for familiarization with application domain knowledge for ML researchers tackling application problems 2. A comparison between an LCAO wave function model of a molecule and a representation of it in a graph convolutional network. 3. An energy-prediction neural network called QDF, based on LCAO evaluated in the atom locations followed by two fully connected neural networks, one predicting energy, one predicting potential in order to encourage a Hohenberg-Kohn-type density-to-potential one-to-one mapping.

Strengths: It is true that the field of chemistry does not benefit from chemistry ML research performed in the vacuum, leading to e.g. molecule generative models making 18-atom aromatic rings. It is good for ML researchers to know the language and objectives of their field of application. The LCAO-based QDF does a good job at predicting molecular properties for atom numbers outside the range of molecule sizes in the training set, compared to a previously published GCN.

Weaknesses: There are worrying imprecisions in many places of the manuscript. LCAO is not really a basic assumption (l38), but a first-order approximation - notably it is the solution to the Born-Oppenheimer-approximated Schrödinger equation at infinite distance between atoms. LCAO being an approximation means that it has limits and requires corrections, which end up being nonlinear. Criticizing GCN for being too nonlinear seems out of place in that regard -- LCAO is not nonlinear enough in the same sense. l57 states that one can conclude from the fact that the Hohenberg-Kohn map is nonlinear that one can model it with a simple feedforward DNN, which is not an adequate conclusion without evidence. l61/62 state that the proposed QDF model competes with SchNet on the qm9 energy prediction task, but no results on full qm9 are listed anywhere. The comparison of LCAO to GCN seems somewhat forced. The proposed model could be presented completely without this comparison and not much information would be lost. The descriptions of the LCAO and QDF network approaches conspicuously lack a harmonic term. Is this actually omitted in this work? The polynomial term is radial so it cannot absorb spherical harmonics into a cartesian polynomial. Omitting spherical harmonics does not do the orbital functions justice and will results in lack of precision. Consider adding these if they are not present. If they are present, why are they omitted in the manuscript? The present work is not the first to show that simple functionals of atom position involving orbital-like functions followed by a simple regression can compete with state-of-the-art neural network approaches for energy prediction. See e.g. [Eikenberg 2018, Solid Harmonic Wavelet Scattering for Predictions of Molecule Properties]. While it is laudable that the present work proposes a new task on QM9, namely to split the dataset according to molecule sizes, it neglects several subtleties. QM9 is called QM9 because it is a dataset of molecules with up to 9 heavy atoms (counting atoms excluding hydrogen). Splitting the dataset at atom count instead of heavy-atom count will lead to a diverse mix of heavy-atom counts being on either side of the split. For the purposes of many property prediction tasks in non-ionized and stable molecules, the hydrogen atoms are subsumed and thus irrelevant for the prediction of the property, so in effect the proposed split does not actually lead to extrapolation. An evaluation of splitting according to heavy atoms was proposed in [Eikenberg 2018] and suggested as a general benchmark. Additionally and unfortunately, the QM9 dataset does not represent all heavy-atom-counts with the same frequency. It would be a good idea to propose several dataset splits and not only one.

Correctness: See weakness. Several statements are imprecise.

Clarity: The paper lapses into imprecision at several points, often under the guise of simplifying things for the ML researcher.

Relation to Prior Work: See weaknesses. Some prior work modeling molecule properties using simple Gaussian-type orbitals followed by regression was not taken into account. It also avoids comparison to previous methods on any of the original QM9 tasks.

Reproducibility: No

Additional Feedback: It is questionable whether the repeated insinuations that ML researchers have no understanding at all about the subject matter they are working on are well-placed in this context. For ML researchers it is generally advisable to be informed about their domain of application and I believe most ML researchers take this to heart. If there is a pervasive feeling that this is not the case for computational chemistry, then it might be a good idea to publish an opinion piece into a visible location. Admittedly, the Neurips community might be exactly the right addressee for this (but a Neurips paper might not be the right vehicle). However, it is entirely unclear to me who exactly is meant by “the ML community” from the point of view of the authors. The authors explicitly exclude [24, 23] (see line 250), which to me are central machine learning works in chemistry. An elucidation of who is meant by “the ML community” would be helpful information. Minor: “Jones” should be cited as “Lennard-Jones”


Review 2

Summary and Contributions: This paper compares different deep learning approaches to modeling the quantum mechanic properties of molecules, and presents a model that incorporates multiple ideas from physics including (1) the linear combination of atomic orbitals to obtain molecular orbitals; (2) a constraint on the external potential function. While none of these ideas are new in themselves, the authors provide a clear and useful analysis describing the differences between modeling approaches (i.e. graph convolution networks and density functional theory models from physics), and provide a particular model that captures important physics intuition and does well in experiments.

Strengths: There are many ways to use deep learning to model data with complex structure, such as graphs and molecules, and understanding the relevant issues is important NeurIPS community, especially consider how much of the work on graph neural networks has been motivated by applications to small molecules. This work attempts to tease apart some of the subtle (but important) differences in these models for a particular physics-oriented application, but I think the lessons are relevant beyond the particular scope of this application. The authors provide compelling theoretical grounding and empirical evaluation.

Weaknesses: The proposed method is only compared to alternatives on a single task.

Correctness: I had no complaints.

Clarity: The paper is clear and well written.

Relation to Prior Work: Yes.

Reproducibility: Yes

Additional Feedback:


Review 3

Summary and Contributions: The paper studies the link between graph convolutional networks (GCNs) trained on molecules for the prediction of molecular energies and the linear combination of atomic orbitals (LCAO) method from quantum chemistry. In doing so, the authors contribute the following: 1. Parallels between the two approaches. 2. Differences between the two approaches, and potential shortcomings of GCNs for molecules. 3. Based on the two previous points, a new method which the authors refer to as quantum deep field (QDF). 4. Empirical evidence of the efficacy of the QDF model based on an extrapolation task, not merely interpolation. 5. The authors also advocate for new benchmarks in machine learning for physical sciences in which models are judged by their extrapolation capabilities, which the authors suggest is a better metric to determine whether the machine learned model learned underlying physical principles or simply fit non-physical patterns within the training set.

Strengths: 1. The link between GCNs and the LCAO method provides insight into what GCNs for molecules get right, and what they get wrong. In light of the referenced work on the difficulty of improving GCN performance with depth, the comparison and subsequent arguments presented by the authors are compelling. For anyone in machine learning designing GCNs specifically with molecules in mind (seemingly a non-trivial subset of graph neural network papers given their use in chemistry and drug design, among other areas), the insights provided in this paper are highly relevant. 2. Based on these arguments, the architecture for the QDF is well motivated. I found particularly interesting the way in which the physical constraints are imposed via the Hohenberg-Kohn theorem/map, which seemed to have significant novelty over standard physical loss constraints in many other papers. 3. The empirical results, while constrained to QM9, are compelling, particularly the extrapolation capabilities of the QDF method compared to the GCN. 4. I am sympathetic to the authors advocacy for judging machine learned models for physical sciences by their extrapolation capability, and think this point is valid for anyone in the NeurIPS community applying machine learning to physical problems.

Weaknesses: 1. In the larger field of GCNs generally, not specifically for molecules, the insights presented in this paper are perhaps of limited interest. 2. I would have liked to see further discussion on the computational complexity of the proposed QDF. While the QDF has significantly fewer learned parameters than, for example, SchNet, how easy/hard is it to train the QDF model and then once trained, to evaluate the QDF on a new molecule? Two particular areas for clarification come to mind: (i) the “alternating” training of the energy DNN versus the HK DNN; and (ii) the computational complexity associated to needing to use a grid with G grid points to evaluate the vector-format atomic orbital.

Correctness: Everything appears to be correct, and the supplementary material provides sufficient detail on the numerical experiments.

Clarity: I found the paper to be very well written. The only exception to this was the last paragraph of Section 2 that contains equation (8). The reasoning for this correct description of the LCAO versus what came before in Section 2 was not quite clear to me, and given that it is an important aspect of the arguments in Section 3.5 some additional clarification would be beneficial. Specifically, why is the dimensionality the same as the number of basis functions, and what do these dimensions represent?

Relation to Prior Work: With regards to GCNs for molecules, the paper very clearly describes how their work compares to such methods, as indeed this is the main point of the paper. However, the QDF method can be more broadly placed in the genre of “machine learning for quantum chemistry,” of which not all methods are GCNs. Comparison to such methods, particularly those which constrain their models through physical considerations (invariants, laws, rules, etc), and in particular the extrapolation capabilities of those models (or lack thereof), would have further strengthened the arguments presented in this paper.

Reproducibility: Yes

Additional Feedback: Regarding the broader impacts, I marked partially. Here are my comments: Broader impacts to drug design, material science, chemistry are briefly mentioned. The broader impact to GCN design for molecules is clearly discussed. Ethical and societal implications are not discussed, but that would seemingly be beyond the scope of this work. UPDATE TO REVIEW: I have read the other reviews and the authors’ feedback, and after much discussion among the reviewers, I am updating my review as follows. Overall my view of the paper is still favorable. Review #2 does raise some good points, though, and the author feedback raises some additional concerns. In particular: * Review #2 pointed out that the extrapolation task is not really chemical extrapolation. To the authors’ credit, they did new numerical experiments during the rebuttal period in which the molecules were split by the number of heavy atoms, not the total number of atoms. The results the authors get are good, however, they are about the same as the results reported in [Eikenberg 2018] (referenced in Reviewer #2’s review) for the same task.
 * The fact that the computational complexity of the method prohibits training on the full QM9 dataset is worrisome for the long term outlook of the proposed method. Indeed, there are competing methods, such as the 2017 paper introducing ANI-1 [https://doi.org/10.1039/C6SC05720A], that are capable of absorbing millions of molecular conformations into their training set, and which are thus capable of learning complex patterns in chemical compound space.
 Having said that, I still think the paper has these positive points: * A novel approach, as illustrated in Figure 1. * An interesting contrast between the proposed approach and GCN approaches. * Compelling numerical results in comparison to a standard GCN approach as illustrated in Figure 4. Even if this task is not chemical extrapolation, it still shows the benefit of the proposed approach over GCNs. * Discussion of extrapolation metrics in the ML for chemistry field, and the desire to be able to learn physical models from data.
 Balancing these considerations, I am revising my overall score from 8 to 7. While this slightly lowers my overall score, I am still solidly in favor of accepting the paper.

[Author Response · NeurIPS 2020]

We thank the reviewers for the thoughtful feedback in these difficult times caused by the global COVID-19 pandemic.

**1. LCAO is not really a basic assumption (l38), but a first-order approximation, and ends up being nonlinear (R2).** We agree that LCAO serves as a first-order approximation; in the revised manuscript, we have corrected line l38 and the other relevant sentences. However, in QM9, the energy is calculated using LCAO (6-31G); therefore, when QM9 is used for training, the model must be based on LCAO, and QDF achieved high extrapolation performance. If the dataset included "experimental data," a nonlinearity would be entailed.

**2. The HK map is nonlinear and modeled with a feedforward DNN, which is not an adequate conclusion without evidence (R2).** The feedforward DNN is a choice for modeling the nonlinearity of the HK map from $\rho$ to $V$; indeed, the HK map has been modeled using kernel methods [23]. As you may be aware, our HK map is modeled in an LDA (not a GGA) fashion. We emphasize that even this LDA-like HK map achieved high extrapolation performance.

**3. Results on full QM9 (R2).** QDF requires substantially more training time and memory than those required by a GCN, because of the large number of GTOs, grid field points, and the alternate learning of two loss functions. Therefore, we could not train QDF on all 130,000 molecules included in QM9 using our current computational environment. Training QDF on full QM9 necessitates the efficient use of dozens of GPUs. We will address this in future work.

**4. The comparison of LCAO to GCN seems somewhat forced. QDF could be presented without this comparison (R2).** Considering the NeurIPS community, it is important to provide the comparison between LCAO and GCN with regard to the recent trend of molecular GCNs. Of course, QDF can be proposed without a comparison to GCN. In fact, we have another work involving the application of QDF to materials informatics that will be published as a physics paper. It would not make much sense to describe LCAO in detail and compare LCAO to GCN in that paper.

**5. Spherical harmonics in GTO (R2).** Thanks for this critical comment. Yes, we omitted the spherical harmonics in the GTO for simplicity. As you have commented, we simplified some descriptions so they can be easily understood by readers in the NeurIPS community who are not familiar with physics and chemistry. We emphasize that even though the simplified GTO was employed, QDF achieved high extrapolation performance. The improvement of the GTO by considering spherical harmonics and other factors is an important future work.

**6. Eikenberg 2018 (R2).** Thank you for bringing this important related study. We did not cite Eikenberg 2018; however, we have already cited some ML studies that use GTOs for modeling the electron density as [21,26]. In the revised manuscript, we have additionally cited Eikenberg 2018 and discuss these related studies in detail.

**7. An extrapolation evaluation of splitting according to heavy atoms (R2).** Thanks for the constructive and beneficial feedback. In this rebuttal period, we split QM9 according to the heavy atoms and evaluated the extrapolation performance. After training the model using data regarding molecules with less than 7 heavy atoms (# of samples is 3,000), the interpolation error is 0.10 eV; further, the extrapolation error for molecules with 8 and 9 heavy atoms (# of samples is 127,000) is 0.19 eV. As you mentioned, QM9 has a substantial bias (97 % of QM9 corresponds to 8 and 9 heavy atoms); however, our QDF was successful in performing the extrapolation.

**8. The phrases of "ML researchers" and "ML community" (R2).** We apologize for using these somewhat confusing phrases. To prevent confusion among readers, we have excluded these phrases from the revised manuscript.

**9. QDF is only compared on a single task (R3).** We used a single benchmark dataset; however, we evaluated the extrapolation performances for three different energy properties in QM9. For predicting other properties, such as the HOMO and LUMO, QDF needs to capture the excited states of molecules, which is different from the ground-state energy and will be more difficult to predict. We will perform model extension in this regard in future work.

**10. The computational complexity of QDF (R4).** As we had mentioned in Response 3, considering the grid field points of all molecules entails a high memory cost. However, once the model is trained (e.g., using a GPU, the model can be trained for 10,000 molecules within 24 hours), the prediction can be performed within a second for a molecule.

**11. Why is the dimensionality the same as the number of basis functions, and what do these dimensions represent (R4)?** An initial assumption in LCAO is that the number of molecular orbitals $N$ is equal to the number of atomic orbitals (or basis functions) in the linear combination. This is why the number of orbitals $N$ corresponds to the dimensionality of the vector on a position $r$, i.e., $\boldsymbol{\psi}(r) \in \mathbb{R}^N$, and this $N$ determines the computational approximation and accuracy. In the revised manuscript, we have clarified this, particularly for readers in the NeurIPS community.

**12. QDF can be more broadly placed in the genre of "ML for quantum chemistry" (R4).** Yes, QDF is an ML model for quantum chemistry and can be viewed as one of the "physics-oriented" approaches, such as the physics-informed, Hamiltonian (graph), and Fermionic neural networks (PINN, HNN, and FNN) [Raissi 2019, Pun 2019, Greydanus 2019, Sanchez-Gonzalez 2019, Pfau 2019]. Among these, the HNN, which was proposed in the NeurIPS community, involves physical considerations in training the neural network and addresses some classical mechanics problems. In that sense, our QDF is also such a neural network, but it focuses on solving quantum chemistry problems.

[Meta-Review · NeurIPS 2020]

The paper compares different deep learning approaches to modeling the quantum mechanic properties of molecules, and presents a model that incorporates multiple ideas from physics. Some reviewers appreciated multiple aspects of the paper, including: - A novel approach, offering an interesting contrast to GCN approaches - Compelling numerical results in comparison to a standard GCN approach (even if this not extrapolation, it still shows the benefit of the proposed approach over GCNs). - Discussion of extrapolation metrics in the ML for chemistry field At the same time, a few questions were raised, including: - certain imprecisions and over-claims - LCAO being first-order approximation rather than "basic assumption". The proposed approach does not even use spherical harmonics in the LCAO representation. The fact that this "still works well" is bothering as it necessarily leads to a bad approximation of the molecular orbitals, to a point where the added benefit of chemically informed modeling might be almost gone. - worrysome computational complexity and insufficient discussion thereof - unacceptable argument to not use spherical harmonics to make the approach "simpler for an ML person to understand" The reviewers read the rebuttal and thoroughly discussed it. Overall, the impression is positive and our recommendation is to accept the paper. It is our hope that the authors revised the paper accounting for detailed and thorough comments provided by the reviewers. The AC also invites the Authors to mild the insinuations that ML researchers have no competence in the application domain, as being arrogant and often not true.